# Robot Learning with Sensorimotor Pre-training

**Ilija Radosavovic  Baifeng Shi  Letian Fu  Ken Goldberg  Trevor Darrell[†]  Jitendra Malik[†]**

University of California, Berkeley

**Abstract:** We present a self-supervised sensorimotor pre-training approach for robotics. Our model, called RPT, is a Transformer that operates on sequences of sensorimotor tokens. Given a sequence of camera images, proprioceptive robot states, and actions, we encode the sequence into tokens, mask out a subset, and train a model to predict the missing content from the rest. We hypothesize that if a robot can predict the masked-out content it will have acquired a good model of the physical world that can enable it to act. RPT is designed to operate on latent visual representations which makes prediction tractable, enables scaling to larger models, and allows fast inference on a real robot. To evaluate our approach, we collected a dataset of 20,000 real-world trajectories over 9 months using a combination of motion planning and grasping algorithms. We find that sensorimotor pre-training consistently outperforms training from scratch, has favorable scaling properties, and enables transfer across different tasks, environments, and robots.

**Keywords:** Robot Learning, Self-supervised, Sensorimotor, Pre-training

## 1 Introduction

Over the last couple of years, inspired by vision [1, 2, 3, 4] and language [5, 6, 7], there has been an increased interest in pre-training for robotics. For example, we have seen promising results from self-supervised visual pre-training on large and diverse image collections [8]. However, robotic data contains rich sensory and motor information that is difficult to capture with visual pre-training alone. We ask: *can we learn good sensorimotor representations from robotic trajectories?*

In this paper, we propose a self-supervised sensorimotor pre-training approach for robotics. We formulate robotic pre-training as a general sensorimotor sequence prediction problem. We instantiate this idea through a masked prediction task, similar to the counterparts in natural language processing and computer vision [6, 3, 4]. We hypothesize that if a robot can predict missing sensorimotor content it will have acquired a good model of the physical world that can enable it to act.

Our model, called RPT, is a Transformer [9] that operates on sequences of sensorimotor tokens. Given an input sequence of camera images, proprioceptive robot states, and actions, we encode the interleaved sequence into tokens, mask out a subset of the sequence, and predict the masked-out content from the rest. We perform masking across all modalities and time using a high masking ratio, which encourages the model to learn cross-modal, spatio-temporal representations.

We encode camera images using a pre-trained vision encoder [8] and use latent visual representations for sensorimotor sequence learning. This enables us to build on strong visual representations, trained on large and diverse image collections from the Internet. Compared to prediction in pixel space, performing prediction in the latent representation space makes the task more tractable. This design decouples the computational vision cost from the sensorimotor context length, making 10 Hz control with over 300M parameter models and large context lengths feasible on a physical robot.

---

[†]Equal contribution. Videos are available on our project page.

7th Conference on Robot Learning (CoRL 2023), Atlanta, USA.

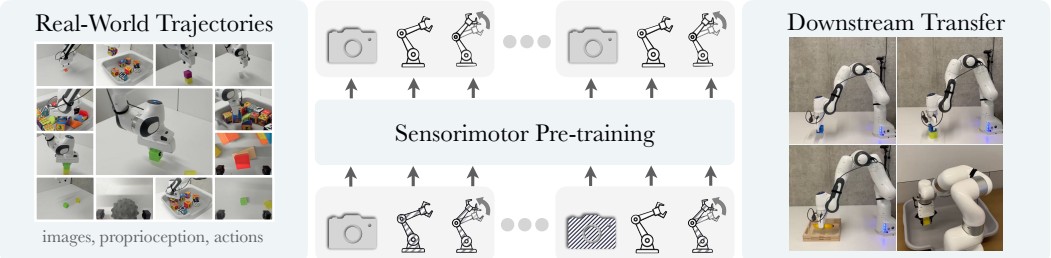

Figure 1: **Robot learning with sensorimotor pre-training.** *Left:* We collect a large dataset of real-world trajectories that contain multiview RGB images, proprioceptive robot states, and actions to use for sensorimotor pre-training. *Middle:* Given a sensorimotor trajectory, we mask out a subset (shown by striped pattern) and train a Transformer model to predict the masked-out content from the rest. *Right:* We transfer the pre-trained representations to different downstream tasks and robots.

To study our pre-training approach, we collected a dataset of over 20,000 real-world trajectories over 9 months using a combination of motion planning and model-based grasping algorithms. Each trajectory is a sequence of multi-view RGB camera images, proprioceptive robot states, and actions. We collected trajectories for classic robotic tasks, namely, single object picking, bin picking, stacking, and destacking. All of the tasks include variations in object pose, shape, and appearance.

To understand the effect of pre-training, we perform a series of real-world experiments. We find that RPT consistently outperforms training from scratch and that the improvements are larger for harder tasks (up to $2\times$ for the block stacking task). We also find that our sensorimotor pre-training approach enables successful transfer across different tasks, lab environments, and robots. Moreover, our approach has favorable scaling properties and benefits from better vision encoders, longer sensorimotor context lengths, and larger pre-training datasets. Finally, we find that masking across both modalities and time, with a high masking ratio, is important for good performance. We encourage the readers to see the the extended version of this work on arXiv and also to check the project page.

## 2 Robot Learning with Sensorimotor Pre-training

Our approach consists of a pre-training and a fine-tuning stage (Figure 1). We pre-train sensorimotor representations with masked prediction on sequences of camera images, proprioceptive states, and actions (Figure 2). After pre-training, we transfer representations to downstream tasks (Figure 3).

### 2.1 Sensorimotor Pre-training

We begin by describing the pre-training stage of our approach. In the pre-training stage, we are given a dataset $\mathcal{D}$ of sensorimotor trajectories $\mathcal{T}$. Each sensorimotor trajectory is a sequence of camera images, proprioceptive robot states, and actions: $\mathcal{T} = (i_1, s_1, a_1, ..., i_T, s_T, a_T)$. We assume no access to additional semantic information, like language instructions or task labels. We hypothesize that the unlabeled sensorimotor trajectories implicitly encode the structure of the physical world and that we can use them to learn general sensorimotor representations for downstream robotic tasks.

We formulate robotic pre-training as a general sensorimotor sequence prediction problem, across all modalities and time. We instantiate this idea through a masked prediction task, similar to the counterparts in vision and language [6, 3, 4]. We mask out a subset of a trajectory and train a model to predict the missing content. Specifically, given a sensorimotor sequence of $L$ tokens, we sample a mask sub-sequence $M \subset [1, L]$ and train a model to minimize the mean squared error of the masked tokens $\mathcal{T}_M$ conditioned on the observed tokens $\mathcal{T}_{[1,L] \setminus M}$. The intuition is that if a robot can infill missing sensorimotor content it will have acquired a good model of the physical world that can enable it to act. This general formulation enables us to represent many different contextual prediction problems by simply using different masking patterns. We consider a number of variants, including random masking at the modality, timestep, and token level, as well as causal masking.

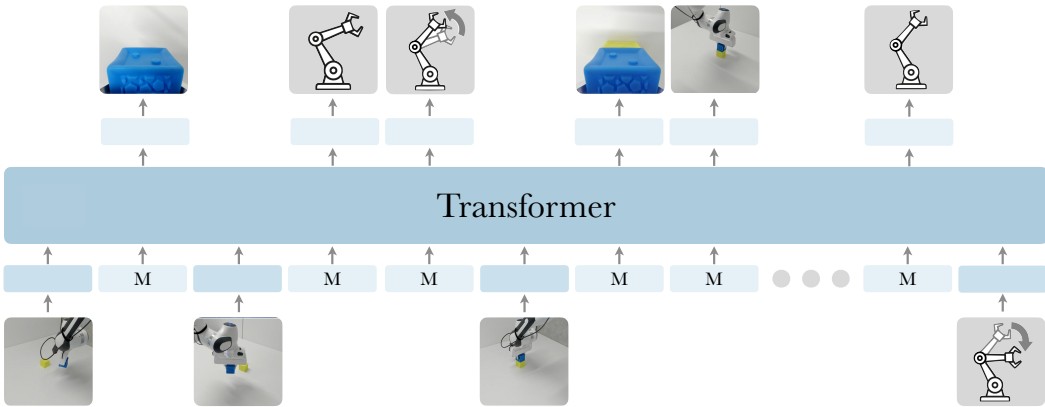

Figure 2: **Sensorimotor pre-training.** Our model is a Transformer that operates on interleaved sequence of camera images, proprioceptive robot states, and past robot actions. We encode sensorimotor inputs into tokens, mask out a subset, and train a model to predict the missing content.

## 2.2 Architecture

**Token masking.** Given a sequence of sensorimotor tokens we mask out a subset. We represent each masked out input using a mask token that is learnt per modality and shared across time. We mask the input tokens independently without taking the time step or modality into consideration which encourages the model to learn to correlate information across both modalities and time.

**Vision latents.** Our sensorimotor model must process sequences of high-dimensional images from multiple views, which is challenging from both the learning and the computational perspective. To overcome this, we use a pre-trained vision encoder to compute visual representations [8] and operate in the latent space. This enables us to build on strong visual representations, trained on large and diverse images collections from the Internet. Compared to prediction in pixel space, prediction in the latent space makes the task more tractable. This design also decouples the computational vision cost from the sensorimotor context length, making fast inference with large models feasible.

**Token encoders.** We use a separate linear encoder per modality. Since our modality encoders do not share weights, we do not use additional modality embeddings. To represent time, we add positional embeddings to each token. All tokens from a single timestep share the positional embedding value.

**Transformer model.** The encoded and masked sequence of tokens is passed into a Transformer model [9]. We follow the standard Transformer design consisting of a series of Transformer blocks with multi-head self-attention operations. Our model predicts latent representations for each input.

**Prediction heads.** We decode the hidden representations into prediction targets using linear project layers. Each latent is decoded from the hidden size back to the original modality dimension. We make predictions in the original input space for joints and the visual latent space for images.

**Masked objective.** We compute the mean squared error reconstruction loss between the predictions and the ground truth input values. We apply the loss only to the predictions from the latent representations corresponding to the masked inputs. Predictions for the observed inputs incur no loss. To weight the importances and scale of different modalities we apply a per-modality loss weight.

## 2.3 Downstream Transfer

Our goal is to learn general sensorimotor representations that can be transferred to different downstream tasks and robots. Inspired by vision and language [6, 3, 4], we explore two settings: *(1) Fine-tuning.* We fine-tune a pre-trained model checkpoint on downstream task data. In this setting pre-training can be seen as providing a good initialization for learning. *(2) Linear probe.* We use a frozen pre-trained model to extract sensorimotor features and train a single linear layer to predict actions. This enables us to evaluate the quality of the pre-trained representations alone.

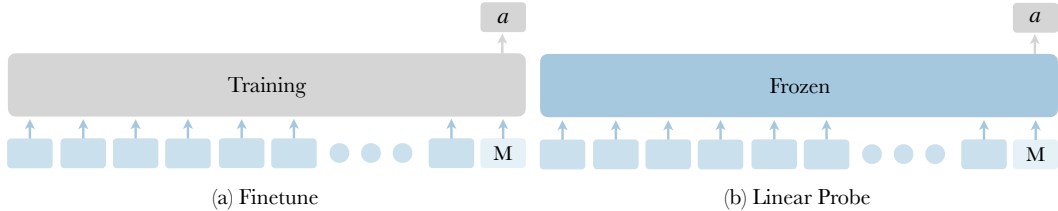

(a) Finetune  (b) Linear Probe

Figure 3: **Downstream transfer.** We consider two different settings for evaluating representations on downstream tasks: (a) *Fine-tuning:* We fine-tune the entire pre-trained model on the downstream task data; (b) *Linear Probe:* We freeze the pre-trained model and train a linear action read out layer.

## 3 Experimental Setup

**Robot and tasks.** We use a 7-DoF Franka robot with the default 1-DoF parallel jaw gripper. We perform joint position control at 10 Hz. We include joint positions and the gripper state in the proprioceptive information. We use three RGB cameras, one attached to the robot hand and two on the sides (Figure 9). We consider four different tasks: Pick, Bin Pick, Destack, and Stack. To study the proposed approach, we collected a dataset of sensorimotor trajectories using a combination of motion planning and model-based grasping algorithms (see Appendix A for more details).

**Vision encoder.** We use the Vision Transformer (ViT) architecture [10] to encode image inputs. Specifically, we use the pre-trained models from [8] which were trained via MAE [4] on a collection of 4.5M images from Ego4D [11], Epic [12], Something-Something [13], 100 Days of Hands [14], and ImageNet [15] images. We extract features from all three cameras using the same model. We use the mean pooling of output tokens from the vision encoder as the vision feature.

**Sensorimotor transformer.** Our sensorimotor model is an encoder-only Transformer with ~1M parameters. The transformer has a hidden dimension of 192 and four transformer blocks, each with 4 heads and an MLP ratio of 2. The sensorimotor input contains multiple steps, each step including the visual features from 3 cameras views, the proprioceptive states and actions. Since the input from different modalities have different dimensions (*e.g.*, 768 for images and 8 for proprioception and actions), we project each modality to the same dimension of 192 using a linear layer per modality.

**Pre-training.** During sensorimotor pre-training, we first randomly sample a masking probability from a fixed range of masking ratios, and then independently mask each input token (visual, proprioception, or action) with the same probability. The default masking ratio range is $[0.7, 0.9]$, which we find to work the best empirically. We also study other masking strategies such as masking all the tokens from a modality or a timestep. Please see ablations on masking strategy and masking ratio in Section 4.5. The masked reconstruction loss for each modality is weighted together using a uniform weight. All models are pre-trained for 300 epochs with a batch size of 4096 and 50 warm-up epochs. We use the AdamW optimizer with a learning rate of $4 \times 10^{-4}$ and a weight decay of 0.01.

**Fine-tuning.** We initialize the sensorimotor model with the pre-trained weights and fine-tune it with behavior cloning on the downstream task. At fine-tuning time, the model takes in a sequence of the same length as in pre-training with the action at the last time step replaced with a mask token. To predict the action, we use the output token that corresponds to the mask input token and discard the other predictions. We train a linear layer to predict next 16 actions from the last mask token. We fine-tune the model for 900 epochs for the task of Stack and 300 epochs for other tasks. Other hyperparameters such as learning rate and batch size stay the same as in pre-training.

**Inference.** When testing the fine-tuned model, we feed the past sensorimotor trajectory as input and the model predicts the actions for the next 16 steps. The predicted actions are passed to the robot controller which executes the actions at 10Hz. We experimented with executing one action at the time and re-predicting but did not observe a significant difference in performance. After each action is executed, the visual observations and the achieved state are recorded and, alongside the executed action, are fed back as the sensorimotor inputs for the next prediction in the autoregressive fashion.

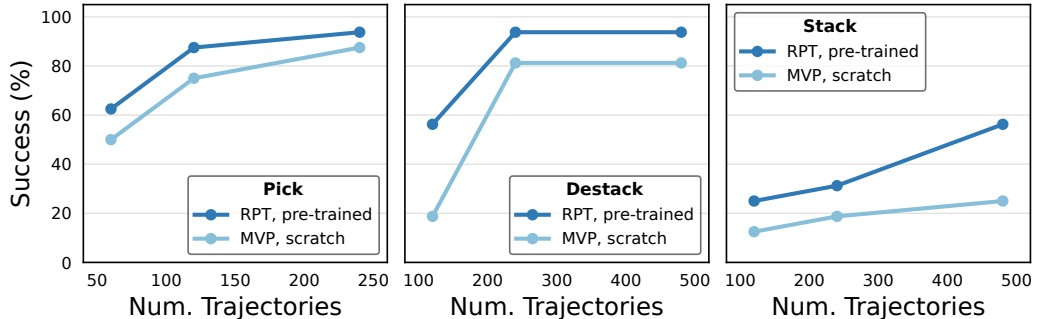

Figure 4: **Sample complexity, fine-tuning.** We study the effect of *sensorimotor pre-training* as the amount of fine-tuning data increases. We find that sensorimotor pre-training (RPT) brings consistent improvements over training the sensorimotor model from scratch (MVP) and that the gains are larger for the harder tasks (Stack). Note that the vision encoders are pre-trained and frozen for both [8].

# 4 Experimental Results

We perform evaluations in the real-world and study sample complexity on different tasks, transfer across tasks and robots, scaling properties, and different design decisions. For all experiments, we report success rates across 16 real-world trials with variations in object positions and orientations.

## 4.1 Sample Complexity

We begin by studying the effect of sensorimotor pre-training by comparing its fine-tuning performances to training form scratch. As our scratch baseline we use an improved version of MVP [8], where the single-step MLP policy is replaced by a multi-step Transformer policy. For fair comparisons, we use the same pre-trained vision encoders, sensorimotor architectures, and optimize the learning rate, batch size, and the number of epochs per model. We consider three downstream tasks of increasing difficulty: picking, destacking, and stacking. We pre-train using a different subset of the data from the same task as in fine-tuning and hold out an unseen set for evaluation. In Figure 4 we show the performance as the number of fine-tuning demonstrations increases. We observe that pre-training leads to consistent improvements over training from scratch, across different tasks and data regimes. Moreover, the improvements are the largest for the hardest block stacking task.

## 4.2 Transfer Across Tasks

In the previous section we used the data from the same task for pre-training and fine-tuning. To evaluate if our pre-training approach can learn general sensorimotor representation instead of representation for a specific task, we study pre-training and fine-tuning across different tasks. Specifically, we first pre-train a model on data from different tasks: stacking, picking, successful bin picking trajectories, and bin picking trajectories that include failure cases. We then fine-tune and evaluate the models on stacking. We report the results in Figure 5. We observe that pre-training on all of stacking, picking, or bin picking lead to similar downstream performance on stacking, which suggests that our sensorimotor pre-training approach can learn transferable representations across tasks. We also see lower performance when pre-training on all of bin picking data that includes failed trajectories, which highlights the importance of pre-training data quality.

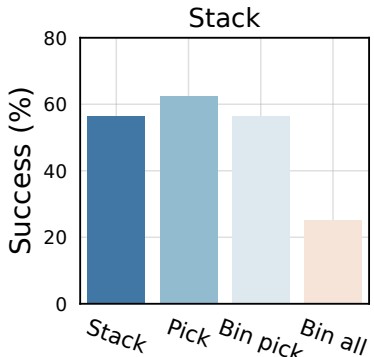

Figure 5: **Transfer across tasks.** We compare pre-training on different tasks and fine-tuning on stacking. We see that pre-training can lead to strong downstream performance even across tasks.

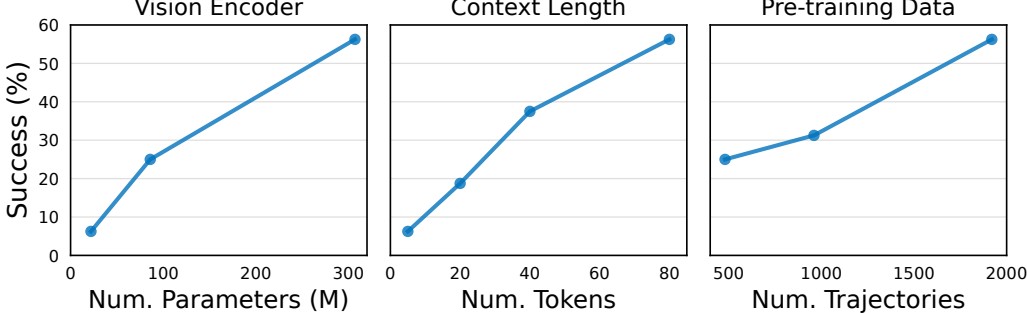

Figure 6: **Scaling studies.** We find that our approach benefits from better vision encoders (left), larger context lengths (middle), and more pre-training data (right). Evaluated on block stacking.

## 4.3 Scaling Studies

**Vision encoder.** We study the performance of our pre-training approach as the size of the vision encoder increases. In all cases, we use a pre-trained and frozen vision encoder from [8] Specifically, we consider three ViT variants of increasing size: ViT-S, ViT-B, and ViT-L. We evaluate performance on the block stacking task which requires precise spatial localization. The results are shown in Figure 6, left. We observe that the performance improves significantly with better vision models. We note that our model is still capable of 10 Hz inference even when using the ViT-L vision encoder.

**Context length.** We compare sensorimotor pre-training with varying context lengths. Namely, we consider context lengths of 1, 4, 8, and 16 timesteps. Note that each timestep contains 5 tokens. In Figure 6, middle, we observe that pre-training on larger contexts leads to consistent improvements, which may suggest that longer contexts may facilitate richer sensorimotor pre-training problems.

**Pre-training data.** We study scaling of our approach as the amount of pre-training data increases. We consider pre-training on 480, 960, and 1920 trajectories and evaluate downstream performance on block stacking. In Figure 6, right, we observe that our approach benefits from more pre-training data which is a promising signal for scaling sensorimotor pre-training to larger trajectory collections.

## 4.4 Transfer Across Robots

**Cross-lab transfer.** We evaluate transfer across different robots. We first consider transfer to a different instance of the same robot type. The downstream robot is in a different lab that has differences in the environmental conditions, varying camera placement, background, and lightning. We compare pre-training to training from scratch. We report the results in Figure 1a and observe that pre-training outperforms training from scratch considerably.

**Cross-robot transfer.** Next, we push this setting further and evaluate transfer across different robot types. Specifically, we pre-train on xArm data from [8] which includes 640 trajectories collected via teleoperation across 8 different tasks. Note that the gap here is quite large, with differences in the robot type, camera type and placement, tasks, background, lightning, data frequency, and collection strategy. We compare pre-training on xArm to (a) no pre-training and (b) pre-training on the same amount of original Franka data. In Table 1b, we see that pre-training on xArm outperforms training from scratch considerably and comes very close to pre-training on the same robot.

| pre-train | fine-tune | success (%) |
|---|---|---|
|  | Franka B | 25.0 |
| Franka A | Franka B | 68.8 |

(a) **Franka → Franka.** Effective transfer from a franka in one lab to another franka in a different lab, with differences in camera positions, background, and lightning.

| pre-train | fine-tune | success (%) |
|---|---|---|
|  | Franka | 25.0 |
| xArm | Franka | 50.0 |
| Franka | Franka | 56.3 |

(b) **xArm → Franka.** We find that sensorimotor pre-training can be effective even across different robot types and come very close to pre-training on the same robot.

Table 1: **Transfer across robots.** We pre-train on data from one robot and evaluate downstream transfer to a different robot. We experiment with both cross-lab and cross-robot transfer.

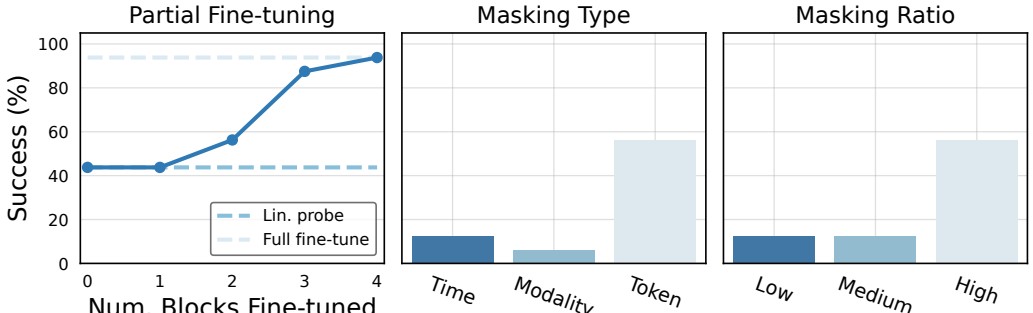

Figure 7: **Ablation studies.** We find that while linear probing achieves non-trivial performance, fine-tuning leads to considerably better results and that fine-tuning a larger portion of the model leads to higher success (left). We observe that masking across all tokens works considerably better than masking all tokens from a timestep or all tokens from a modality at a time (middle). Moreover, using a high masking ratio is essential for learning good sensorimotor representations (right).

## 4.5 Ablation Studies

**Linear probe.** We evaluate pre-trained sensorimotor representations via linear probing. Namely, we extract representations using a frozen sensorimotor model and train a linear layer on top using 120 trajectories for picking. We report the results in Figure 7, left. We observe that linear probing reaches a success rate of $43.75\%$ which is non-trivial but considerably lower than $93.8\%$ with fine-tuning.

**Partial fine-tuning.** In fine-tuning evaluations we fine-tune the full model by default. Here, we study the impact of fine-tuning an increasing number of Transformer blocks. We report the results in Figure 7, left. Note that fine-tuning zero blocks and four blocks is equivalent to linear probing and full fine-tuning, respectively. We observe that the performance increases with the number of blocks fine-tuned and that achieving best performance requires fine-tuning a majority of the blocks.

**Masking type.** We experiment with different masking strategies (see also Section 4.8). Specifically, we consider time-step masking which masks all tokens from a time step, modality masking which masks all of the tokens from a modality, and token masking which masks across all tokens independently. In Figure 7, middle, we see that token masking outperforms the alternatives considerably.

**Masking ratio.** We ablate different values of the masking ratio. As shown in Figure 7, right, there is a significant performance drop when masking ratio is low $[0.1, 0.9]$ or medium $[0.4, 0.9]$ compared to a high masking ratio $[0.7, 0.9]$. This suggests a high redundancy between different time steps and modalities in the input sensorimotor sequences and a high masking ratio is crucial for learning useful representations for downstream tasks. This is consistent with prior work on visual pre-training [4].

## 4.6 Inference Speed

Our sensorimotor model is designed to operate on latent visual representations that are computed per image, which decouples the vision model from the sensorimotor model and makes the vision computation cost independent of the context length. This enables us to use large vision models (up to ViT-L with 307M parameters) with system level inference speed of 10Hz on a 2080 Ti GPU.

## 4.7 Emergent Self-Correction

We observe that some of our models that are pre-trained with sensorimotor prediction can exhibit an emergent self-correction behavior at test time. For example, if the robot fails to grasp an object initially, it would move back, and proceed to grasp the object successfully. Please see the project page for videos. Since these models were pre-trained and fine-tuned only with successful trajectories, this type of behaviors have not been seen during training. Moreover, some of the states were not present in the data at all (e.g., starting a grasp or moving up with a fully closed gripper).

## 4.8 Causal Masking

We experiment with a variant of our sensorimotor pre-training approach that uses causal masking instead of random masking, like in the experiments so far. We make a minimal change and simply sort the sampled mask sequences. Note that the self-attention is still bidirectional rather than causal. The pre-training prediction problem is causal however. We consider two settings: pre-training on the same task (pick to pick) and pre-training on a different task (pick to stack). We report the results in Figure 8. We find that causal pre-training leads to better performance on the same task and worse transfer across different tasks.

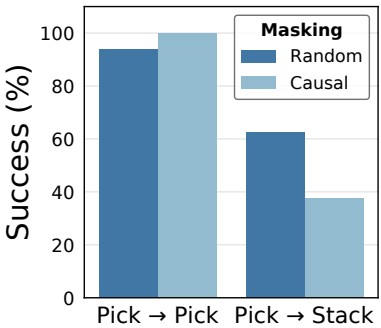

Figure 8: **Causal masking.**

## 5   Related Work

**Self-supervised learning in robotics.** There is a rich body of work on self-supervised learning in robotics. [16, 17] learn grasping policies from large-scale self-supervision; [18] learns visuomotor policies with self-supervised visual correspondence; [19] calibrates learned grasping policies for particular objects based on grasp success feedback; [20] simultaneously collects robot trajectories for multiple tasks and learns corresponding policies; [21] uses an auxiliary contrastive learning task on human demonstrations; [22] learns goal-conditioned robot policies from teleoperation data; [23] uses multi-view reconstruction for model learning. Overall, prior work has largely focused on using self-supervision for learning a particular robotic task. In contrast, we use self-supervision for pre-training general sensorimotor representations that can be transferred to different downstream tasks.

**Multi-task and large models for robotics.** A number of works have explored learning multi-task and large models for robotics. BC-Z [24] performs large-scale vision-based imitation learning; [25] trains policies with cross-domain datasets; [26] learns multi-task Transformer policies for manipulation with language; [27] pre-trains with offline reinforcement learning. [28] trains a generalist agent on trajectories from different tasks jointly. In contrast, we use a masked objective, operate on latent visual representations, and focus on real-world trajectories. [29] trains language-conditioned Transformer policies from human or expert demonstrations. Likewise, we leverage Transformer models but focus on self-supervised learning from sensorimotor trajectories. [30] grounds language models with visual inputs via an embodied visual question answering model. Overall, we share the goal of multi-task robot learning and training large models for robotics but propose a general self-supervised pre-training approach that learns from real-world sensorimotor trajectories alone.

## 6   Discussion

**Limitations.**   We note that our work has several important limitations. First, our dataset is collected using a single robot in a single lab, which limits the diversity and realism of the data. Scaling to more diverse environments and robots remains an important area for future research. Second, the tasks we consider, in both pre-training and fine-tuning, are relatively simple variations of pick and place. It would be good to explore more dexterous tasks with complex dynamics and contacts. Next, our model is pre-trained with the MSE loss and may struggle with multimodality. This is partially alleviated with conditional prediction over large context lengths. Nevertheless, it would be good to explore a generative model variant. Finally, please see the project page for videos of failure cases.

**Conclusion.** We describe an approach for robot learning with sensorimotor pre-training. Our model is a Transformer that operates on sequences of sensorimotor tokens. We pre-train our model by masked prediction. We find that pre-training on this data consistently outperforms training from scratch, leads to $2\times$ improvements in the block stacking task, and has favorable scaling properties. Finally, we demonstrate successful transfer across different tasks, lab environments, and robots.

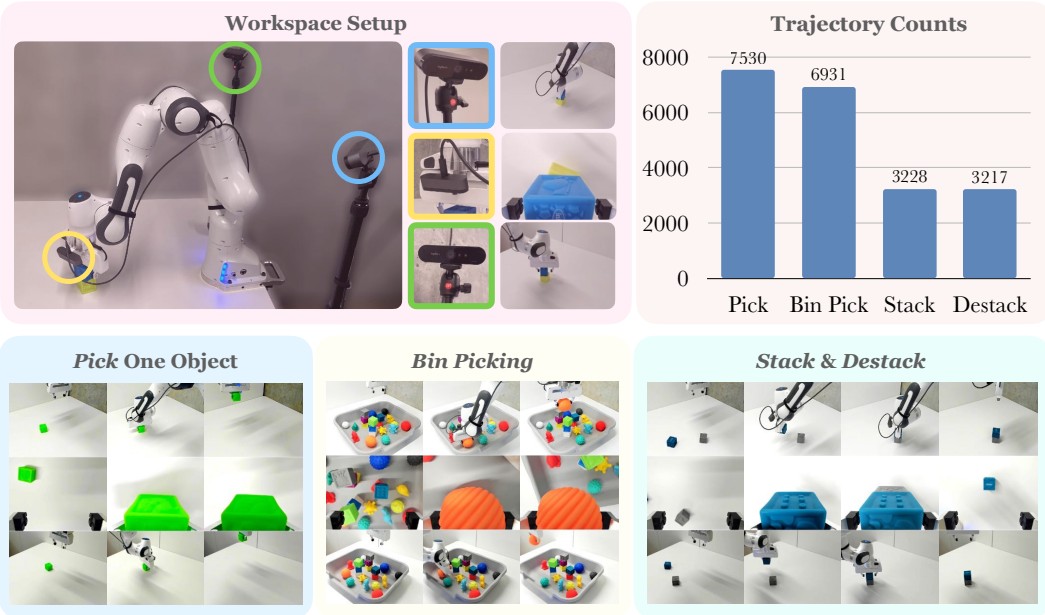

Figure 9: **Dataset.** We collected a dataset of over 20,000 real-world trajectories using a Franka robotic arm. Each trajectory is a sequence of high-quality images from three cameras (one attached to the hand and two on the sides), proprioceptive robot states, and actions. We consider picking, bin bicking, destacking, and stacking tasks, with variations in object position, shape, and appearance.

## Appendix A: Data Collection

**Hardware.** We used a Franka robot with a 7-DOF arm and a parallel jaw gripper (Figure 9, top left). We recorded proprioceptive information in the form of joint positions and velocities. The workspace is equipped with three high-quality Logitech Brio color cameras. One egocentric camera is attached to the robot hand, and two exocentric cameras are attached to the left and the right side of the robot. We synchronized the data streams from the cameras and the robot at 60 Hz and saved data at 30 Hz.

*(1) Pick:* The task is to grasp and lift one out of K cubes from the table. We script a data generation scheme where the robot takes in K cubes at randomly generated 3 DoF poses in robot frame, that are not in collision, and generates sequences of 3 DoF picks-and-places poses for each of the cubes. Then a scripted policy grasps the cube. After grasping, the robot places the grasped cube in the next randomly generated pose, moves back to the start joint configuration, and starts the next grasp.

*(2) Bin Pick:* The robot picks an object from a bin filled with randomly-placed, both soft and rigid, objects (see Figure 9). We use [31] to generate grasps from depth images, captured by an overhead depth camera. Before each grasp, the robot moves to a joint confguration that does not occlude the camera view of the bin. After we obtain a grasp pose, we ise a scripted policy to pick the object. If the gripper is not fully closed the end of the trajectory, we consider the trajectory successful.

*(3) Stack and (4) Destack:* Starting with two cubes of different colors resting on the table, the robot is tasked to first move one of the cubes onto the other cube, and then move the stacked cube back onto the table. The cube poses are generated in a similar fashion as (1). For stacking, the height of the place pose is offset by the height of the cube; similarly, for destacking, the pick pose height is offset by the height of the cube. A scripted policy then executes the pick-and-place trajectory.

**Statistics.** We collected ~20,000 real-world trajectories over the course of 9 months (Figure 9). The dataset includes ~7,000 trajectories for both single object and bin picking, as well as ~3,000 trajectories for stacking and destacking each. The average length of picking and destacking trajectories is ~300 while the stacking trajectories are longer at ~600 steps. The dataset contains variations in object poses, shape, and appearance. We show frames from example trajectories in Figure 9.

**Acknowledgments**

We thank the anonymous reviewers for helpful feedback and suggestions during the review process. This work was supported in part by DARPA Machine Common Sense program, ONR MURI program (N00014-21-1-2801), NVIDIA, Autodesk, and BAIR's industrial alliance programs.

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
