# OpenReview forum: "Robot Learning with Sensorimotor Pre-training"
_robot-learning.org/CoRL/2023/Conference — CoRL 2023 Oral_

### Official Review · Reviewer_8qUw · 2023-07-20

**Confidence:** 5
**Originality:** Fair
**Technical Quality:** Good
**Clarity Of Presentation:** Good
**Impact:** 3

**Recommendation:**

Weak Accept: I recommend accepting the paper, but will not argue for my recommendation if the majority of other reviewers have a different opinion.

**Review:**

Some results are straightforward:
1. Pretraining overall helps.
2. Masking tokens randomly rather than removing a modality / timestep is better
3. Pretraining on negatives makes the model worse
4. Performance improves with number of demos
5. Larger network better

Some results are weird:
1. They show that pretraining on stacking and finetuning on stacking has the same performance on stacking as pretraining on picking/bin-picking and fine tuning on stacking. In fact the former is lower than with pick pretraining. This shows that pretraining is not very relevant, because one would've expected that pretraining on the domain that is closest would've a higher success rate. This discrepancy is neither addressed nor evaluated.
2. Linear probing, ie keeping the pretrained network frozen and with linear action read out gets 40% success. This is surprisingly high and looks fishy.

Some results need more explanation:
1. About speed of 10hz. VIT is large, so how is the image network and sensorimotor model decoupled?

**Quality Of The Limitations Section:**

Limitations are not well addressed

**Questions For Rebuttal:**

1. There is barely any limitations or conclusion. Please write this sincerely.
2. About speed of 10hz. VIT is large, so how is the image network and sensorimotor model decoupled? The decoupling leading to speed increase is not explained well
3. Training procedure is not fully explained. What is the exact loss function used for reconstruction?
4. How many tokens are there per time step? What fraction is the image?
5. Tokenization scheme for non image inputs is not explained

**Robotics Focus:**

Sufficient demonstration on hardware

**Summary Of Paper:**

The paper proposes a network that is transformer based on robotics data. They show that pretraining improves performance over non-pretraining. Then the paper has many ablations on design choices: masking during training, pretraining on specific tasks, context length, etc. The network itself is a pretrained ViT + sensorimotor network.

**Summary Of Recommendation:**

Overall, the paper needs some work, in explaining the network, tokenization schemes, and speed related claims. The results are not necessarily very surprising. It would be better understood when contrasted against previous work like RT-1 / PerAct. Limitations and conclusion is written lazily. The main utility of the paper is not well explained or is not readily apparent. However, though, the paper does put in a lot of work into collecting data and experiments.

---

### Official Review · Reviewer_oCTA · 2023-07-22

**Confidence:** 3
**Originality:** Good
**Technical Quality:** Very Good
**Clarity Of Presentation:** Good
**Impact:** 4

**Recommendation:**

Weak Accept: I recommend accepting the paper, but will not argue for my recommendation if the majority of other reviewers have a different opinion.

**Review:**

### Quality
#### Strengths
- Interesting idea of random maskouts of various modalities
- Experimental setup is thorough - transfer experiments are well-designed given the paper's central claims, and ablations provide insights into the nature of the method
- Experimental results validate the approach.

#### Weaknesses
- Not clear how much this helps with real data, which is limited to the image modality (as pointed out by the paper). We see ablation results showing that masking out all of one modality is very harmful. We don't see an ablation with large stretches of one or two modalities masked out, but I suspect it would behave more like a total lack of that modality (at least at times) than like the random-t random-modality masking - interpolation would be much harder than with the nicely distributed masking.
- Some results, such as the unsuccessful trajectories, would be useful to see
- In general, more results would be appreciated

### Clarity

#### Strengths
- Figure 4: interesting and useful!
- Figures are super stylish
- Writing is overall fine, just verbose

#### Weaknesses
- Intro is difficult to understand - it is heavy in prose, and goes into detail about visual representations, the dataset, and the experiments in uninterrupted paragraph form. This does give *some* sense of what to pay attention to, but readers may benefit more from a clear contribution list
- Figure 1 is a good high-level description, but the middle panel isn't very effective - it's hard to see and the message is nonobvious. After several looks, it seems to be communicating that the input sequence is heterogenous in terms of which modalities appear, but the reconstructions are homogenous and complete in terms of modalities. This can probably be communicated in a simpler way
- Figure 2 doesn't contain as much information as it could. What is the output of the transformer? We know it's a representation, but it may as well be labeled. Furthermore, the mask (a single concept) is indicated three ways - a fade, an "M", and a lighter blue. Not only is this poor data visualization, but the fade is hard to see as a clearly distinguishing factor in the bottom row. The fade also makes it harder to realize that each timestep *has* all three modalities but also may have some masked out.
- Figure 3: low-info and redundant. Feel free to cut.

### Significance
- Transfer performance is promising, even though more results would be helpful. If successful, this capability seems to have robustness potential, making it a significant idea.

### Originality
- I am not aware of any other paper that takes this approach to masking data and reconstructing, then using the learned representations for transfer.

**Quality Of The Limitations Section:**

Additional details required

**Questions For Rebuttal:**

- Why is it exactly that the vision latents from the pretrained vision model allow us to leverage *more* data? Is it because the latents make for a lower dimensional prediction problem and therefore less training time? Is there evidence for this?
- When you say - pretrain the model on real-world robot trajectories "with masked prediction", what do you mean by saying "pretrain on data with masked prediction"? Should this say "masking", or did I misunderstand?
- What's the reason for the double-randomness of randomly sampling a masking probability, then masking with that probability?


**Robotics Focus:**

Highly relevant to robotics but no hardware experiments

**Summary Of Paper:**

This paper presents a method for handling heterogenous data where each modality may not exist at every time step. It hypothesizes that if a robot can predict masked-out regions from a trajectory, it has acquired a good representation of the physical world that can be used downstream.

Methodology consists of doing the following to a demonstration: getting vision latents from the images from a pretrained model; generating an input sequence of these latents proprioception, and action; encoding each modality separately; masking out a subset of input modalities randomly at each time step; and training a model to predict the masked inputs.

The experiment section compares this method to pretraining from scratch in terms of performance and efficiency. Then, the model is used on other data to show transfer capabilities of its learned representations.

**Summary Of Recommendation:**

The novel and intuitive idea is interesting to explore. The transfer results are especially promising, which is what pushes this to an accept. More data would be useful, as well as more detailed and informative statistics and analyses.

Clarity needs various work - see review.

---

### Official Review · Reviewer_Q5ex · 2023-07-22

**Confidence:** 3
**Originality:** Good
**Technical Quality:** Good
**Clarity Of Presentation:** Very Good
**Impact:** 4

**Recommendation:**

Strong Accept: I recommend accepting the paper and will argue for my recommendation even if other reviewers hold a different opinion.

**Review:**

The paper is interesting, well written and timely. I enjoy the premise of the paper, which is to see whether good representation for robotics can be learned from sensory trajectories alone.

I believe the various designs in the paper is or will become standard practices in robot learning, such as using a pre-trained vision encoder and using the latent visual representation for sequence learning, mask reconstruction task as an auxiliary loss. The experimental results are also well thought out. I particularly enjoy the study presented in section 6.2 about the generalizability to different training data. I also enjoy the ablation study on masking type. It is interesting that both timestamp and modality masking are important to obtain good results.

I am recommending acceptance because I think the contribution is relevant to robot learning and I do not find any obvious issue with the paper.

I would like to suggest a small minor correction about the infront speed comparison with RT1. The three hurts in front speed describe when the RT1 paper is system level latency which takes into account camera and action execution. Is the number 10 herd set the author referring to system latency or model inference latency?

**Quality Of The Limitations Section:**

Additional details required

**Questions For Rebuttal:**

Can you also please illustrate why fine tuning is better than linear probe. I understand that this is a phenomena that is also observed in using masked prediction objectives in other research areas, but I would like to understand why this is the case in robotic manipulation.

It will be quite beneficial to the community if the authors release the code and the data sets that accompany the paper. I am happy to raise my score if the authors do so.

**Robotics Focus:**

Sufficient demonstration on hardware

**Summary Of Paper:**

The paper proposes to use masked prediction objectives to learn good representation from images, robot states and past actions. The paper demonstrates that doing so leads to representation that supports more efficient fine-tuning, especially on harder tasks.

**Summary Of Recommendation:**

The paper is interesting, well written and timely. I am recommending accept because I think the contribution is relevant to robot learning and I do not find any obvious issue with the paper. I am happy to raise my scores if the authors provide more analysis and open-source data and code.

---

### Official Review · Reviewer_6uLm · 2023-07-23

**Confidence:** 4
**Originality:** Good
**Technical Quality:** Fair
**Clarity Of Presentation:** Very Good
**Impact:** 3

**Recommendation:**

Strong Accept: I recommend accepting the paper and will argue for my recommendation even if other reviewers hold a different opinion.

**Review:**

## Strengths

This paper investigates an interesting research question; one that is extremely relevant to the robot learning community -- "can representations for robotic manipulation be learned from sensorimotor pretraining alone"?

The paper is quite easy to follow and understand (for the most part -- see "weaknesses" and "minor remarks" below for more). The motivation behind proposing the RPT scheme is clearly established; the model architecture and training scheme are explained in a very accessible manner.

The collected dataset, with its ~20K robot trajectories, is perhaps the key to making RPT work. (If this can be released, it would add value to the paper)

There are some ablation analyses to substantiate the design choices in RPT.

Overall, this work is heading in an interesting direction towards learning general-purpose sensorimotor representations for robotics.



## Weaknesses

In its current form, I have several concerns with the manuscript that I hope to see addressed over the author response phase. My support for this paper would substantially improve if some of these issues are addressed (listed in a partially-ordered sequence of decreasing importance).


[W1] **Comparisons with prior art**: The comparisons made in this work are solely across variants of model architectures, training data, and masking strategies proposed in the paper (i.e., across ViT-B and ViT-L, varying context lengths, eval modes, pretraining tasks).

* However, it is unclear from this evaluation alone as to how RPT compares with other alternative model architectural proposals that have been around for the past year. In terms of masking pre-training objectives for multimodal data, approaches such as MultiMAE (ECCV 2022) [Ref 1] and pre-cursors [Ref 2, Ref 3]. The premise for my assessment is that architectures and masked objectives for modeling multimodal data are neither new, nor the core focus of this paper. And I'd deem it completely acceptable to skip direct evaluation wrt [Ref 1, Ref 2, Ref 3] (I point these out here to encourage a discussion of the nuances of the proposed approach w.r.t these alternative ones).

* In terms of quantitative comparisons, however, the closest set of baselines in the robotics community would be the (extremely broad) class of vision-based imitation learning algorithms. Particularly, for the task-based version of RPT, these baselines (minimally, behavior cloning (BC) ones) need to be evaluated against; to justify the impact of the proposed pretraining architecture. The proposed approach looks like it could exhibit similar shortcomings to vanilla BC approaches, in that once an output action steers the system towards an unknown state, the probability of recovering reduces exponentially for every subsequent action taken.

* Another aspect with the manuscript in its current form is it feels incomplete, as the claims of data-scale, context length etc (Fig. 7) don't yet seem to be saturated. If larger ViT backbones, such as ViT-H / ViT-G, were to be evaluated; the claims would be better substantiated.


[W2] **Generalization**: One of the key claims in this paper is that representations learned by RPT generalize better -- however, I feel like the parameters along with generalization are measured in the current experiment set are somewhat restrictive. I would like a clarification during the author response phase; but I presume all of the data currently collected is from a single Franka Panda arm in a tabletop-style environment? Meaning the extroceptive (images) and proprioceptive inputs (robot states) will implicitly be constrained to work within this robot-environment combination? The representation learning scheme used in RPT does not take a downstream task into consideration (when training on data from across multiple tasks), which means RPT will require to be finetuned when being applied to any downstream task for the first time. I have a number of follow-up questions:

* I see one potential confounding factor in the experimental setup for Fig. 6, which evaluates generalization across various pre-training data subsets. The claim here seems to be that, regardless of the pretraining task (stack, pick, bin pick, bin all), the performance on the downstream task (stack) is fairly consistent (at least for the stack, pick, and bin pick training sub sets). However, one key detail is that regardless of the pretraining data subset, all these models must be finetuned for stacking, using trajectories collected for stacking. This makes it very difficult to assess whether it is the representation itself, or the finetuning phase, which results in performance being on par with a model trained for stacking. (also, there seems to be no clear way to even measure cross-task generalization in the current RPT train-eval framework).

* What happens if either the robot or one of the cameras is moved from the training to evaluation phase (assuming the eventual hand-eye calibration is still accurately available)? (as an aside, it could be interesting for future editions of this work to explore equivariant representation learning)

* Would RPT generalize to a different robot, for the same task? (at least, different instances of the same robot; i.e., another Franka)? Would RPT generalize if one of the joints were constrained to be in a specific pose (i.e., same robot -- but a different set of pose constraints at inference time)? Would RPT work if the environment undergoes major visual changes (while preserving the robot and camera poses)?


[W3] **Missing details; statistical significance**: All experimental studies have been conducted only on 16 trials, which leads to questions about the statistical significance of the presented results. This would, in part, be alleviated if more details about the training and test setup and data became available -- i.e., how different are the train/test tasks in terms of complexity, visual similarity, successful trajectories? There are a number of other detail that I could not find in the paper, such as the dimensionality of the state/action space, the type of normalization or preprocessing applied to the input proprioceptive and controls, any attempts to aid in-/equi-variance of the learned representations. More importantly, it was unclear from the paper as to how RPT is applied at inference time. My understanding was that, upon task-specific finetuning, RPT is run on the input images and robot states to produce robot controls (all other outputs are discarded). However, it is unclear how these outputs interface with the robot controller, workspace constraints; and the rate at which these actions are executed (and, if at all, whether these actions are fed back into the model). This needs to be clearly explained in a revision (and I'd prefer finding space for this in the paper, opposed to the supplementary materials).


### References
----------

[Ref 1] MultiMAE: Multi-modal Multi-task Masked Autoencoders. ECCV 2022.

[Ref 2] LXMERT: Learning Cross-Modality Encoder Representations from Transformers. EMNLP 2019.

[Ref 3] VL-BERT: Pre-training of Generic Visual-Linguistic Representations. ICLR 2020.



## Minor remarks

These comments do not affect my final score; as I believe they are easily addressed in a minor revision. The authors needn't respond to these.

* Line 16: "few year" -> "few years"
* Line 66: incomplete/grammatically-inconsistent sentence
* Line 76: "perspective" -> "perspectives"
* Lines 80-81: The terms "vision cost" and "sensorimotor cost" are not well-defined, and are ambiguious. It would be nicer if they are replaced with terms that are precise. (i.e., "cost" in terms of what? parameteric complexity? space/time complexity? sample complexity?)
* Line 228: sentence needs to be revised for language use / grammar

* I could not make sense of Figure 2 until after I read through sections 3 and 4. I wonder if the figure design may be modified so it is self-contained. (This isn't necessary; but will make the exposition more accessible).
* There is a possibly unintended overloading of the term "fine-tuning" across the paper. The term fine-tuning seems to refer to either (a) training a network from scratch vs fine-tuning a pretrained encoder (e.g., a ViT backbone) for a task (as in Fig. 5), or (b) fine-tuning vs linear-probing the embeddings of a trained/fine-tuned encoder. It would be nice if these two usages had different terms altogether.



# Post-rebuttal update

Based on the convincing rebuttal (see disucssion and comments for more info), the paper has improved significantly. I revise my initial rating of a "weak reject" to a "strong accept". Many thanks to the authors for carrying out these additional experiments to address all reviewer concerns.

**Quality Of The Limitations Section:**

Additional details required

**Questions For Rebuttal:**

I would like to see weaknesses [W1], [W2], [W3] (from above) discussed during the author response and revision phase.

**Robotics Focus:**

Sufficient demonstration on hardware

**Summary Of Paper:**

This paper presents a representation learning scheme (model architecture, training recipe, objective function) for robot manipulation tasks. The model, referred to as RPT (Robotic Pre-trained Transformer), takes as input sequences of images, robot actions, and robot states. At training time, some of these inputs are masked out, and the RPT model is supervised (via a mean-squared error objective) to predict these masked inputs. Once trained, the model may be finetuned for a particlar task, to output actions given an image and robot state. RPT is demonstrated on four manipulation tasks (pick, bin pick, stack, destack) -- evaluated on 16 real-world trials per task (per each experimental study).

**Summary Of Recommendation:**

I feel this paper moves towards a new and exciting direction in terms of representation learning for robotics. My major concern at this point is that, the paper has a number of claims that are not well-substantiated. Addressing this will need multiple additional experiments and analyses that require a major revision. I am open to revising my score based on the author response and discussion, but based on the current version of the manuscript alone (and the fact that there weren't additional supplemental materials to refer to for clarification), I will for now score this a "weak reject".

---

### Author Response · Authors · 2023-08-10
**Summary of Authors' Response**

We thank the reviewers for their time and effort spent on providing careful reviews.

We uploaded a revised manuscript with an expanded limitations section and additional details with the responses to each reviewer individually below.

**We performed the additional experiments requested by the reviewers:**
- [6uLm] Transfer from xArm robot to Franka robot
- [6uLm] Transfer from Franka to different Franka in a different lab
- [6uLm] Transfer to human demonstrations
- [6uLm] Comparison to prior work on visual imitation
- [6uLm] Comparison to prior work on multimodal learning
- [Q5ex, 8qUw] Additional ablations on fine-tuning and linear probe

**Please also check our [supplementary website](https://corl2023-paper476.github.io/), which contains:**
- Animated figures for training and inference
- Example videos of training data
- Videos of trained policies
- Videos of failure cases

**We commit to release all of the code, data, and models to accompany the paper.**

---

### Decision · Program_Chairs · 2023-08-30

**Decision:**

Accept (Oral)

**Comment:**

Based on the original submission, the rebuttal by the authors and the discussion that followed, the reviewers agree that this paper should be accepted to CoRL.
To the authors: great job on writing a strong rebuttal and engaging with the reviewers during the discussion period! This was one of the strongest rebuttals in my batch and I'm really pleased to see the reviewers and the authors engage in discussion that resulted in a much better submission. Please address the remaining comments in the camera-ready version of the paper.